# Gut Microbiota as the Link between Elevated BCAA Serum Levels and Insulin Resistance

**DOI:** 10.3390/biom11101414

**Published:** 2021-09-28

**Authors:** Jan Gojda, Monika Cahova

**Affiliations:** 1Department of Internal Medicine, Kralovske Vinohrady University Hospital and Third Faculty of Medicine, Charles University, 10000 Prague, Czech Republic; jan.gojda@lf3.cuni.cz; 2Institute for Clinical and Experimental Medicine, 14021 Prague, Czech Republic

**Keywords:** branched-chain amino acids, insulin resistance, type 2 diabetes, gut microbiome, gut metabolome

## Abstract

The microbiota-harboring human gut is an exquisitely active ecosystem that has evolved in a constant symbiosis with the human host. It produces numerous compounds depending on its metabolic capacity and substrates availability. Diet is the major source of the substrates that are metabolized to end-products, further serving as signal molecules in the microbiota-host cross-talk. Among these signal molecules, branched-chain amino acids (BCAAs) has gained significant scientific attention. BCAAs are abundant in animal-based dietary sources; they are both produced and degraded by gut microbiota and the host circulating levels are associated with the risk of type 2 diabetes. This review aims to summarize the current knowledge on the complex relationship between gut microbiota and its functional capacity to handle BCAAs as well as the host BCAA metabolism in insulin resistance development. Targeting gut microbiota BCAA metabolism with a dietary modulation could represent a promising approach in the prevention and treatment of insulin resistance related states, such as obesity and diabetes.

## 1. Introduction

Intestinal microbiota composition and function are hypothesized to play a pivotal role in the development of major non-communicable diseases including type 2 diabetes mellitus (T2D), chronic kidney disease, neurodegenerative diseases, and arterial hypertension [1]. The outbreak of the interest in microbiome and its relationship to human health was enabled by the rapid development of the next genome sequencing (NGS) technology in the late 1990s and the term “microbiota” was conceptualized in 2001 [2]. The first studies focused mainly on the microbiota composition but soon it became evident that the products of the microbial metabolism are at least as important as the bacteria inhabiting specific niches of the human body. The way that the intestinal microbiota processes diet components impacts the availability of biologically active end-products. Many of these compounds convey signaling functions as they mediate host–microbiota crosstalk and substantially affect the host physiology. For example, dietary fiber polysaccharides that cannot be cleaved by human glycosidases are fermented by gut microbiota to form short-chain fatty acids (SCFA) that exert pleiotropic effects on the host. They serve as the main energy source for colonocytes [3]; contribute to the maintenance of intestinal barrier [4]; stimulate the maturation of gut-associated immune cells with anti-inflammatory potential [5]; or regulate energy expenditure in the liver, adipose tissue, or skeletal muscle [6]. Among other well-studied end-products are choline, an essential nutrient which is utilized by gut microbes and its deficiency leads to the NASH-like syndrome [7], or acetaldehyde derived from ethanol by intestinal microbiota, which has been shown to disrupt tight junctions between intestinal mucosal cells [8].

Among other emerging microbiota-derived compounds, branched-chain amino acids (BCAAs) have recently gained a significant scientific attention. Indeed, there has been a long-lasting strand of evidence that elevated circulating BCAAs are associated with insulin resistance states such as obesity and diabetes [9] and may even predict cancer development [10,11]. Compared to mammals that are dependent on external supply of BCAAs, numerous gut bacteria possess metabolic pathways allowing for the BCAA biosynthesis. Therefore, it is reasonable to hypothesize that the composition and metabolic performance of intestinal microbiota contribute to the BCAA availability to the host and are, at least partially, responsible for their effect on the host metabolism.

The purpose of the narrative review is to summarize the current knowledge on BCAA body metabolism with respect to the relationship between the BCAA and insulin resistance (IR) and to provide evidence that intestinal microbiota metabolic activity may represent the potential mechanistic link between the diet composition, BCAA availability, and IR development.

## 2. Branched-Chain Amino Acids Are Both Nutrients and Signaling Molecules

BCAAs circulate in plasma as free amino acids (AAs) and are being resorbed virtually to all tissues using specific carriers. BCAAs are unique among other AA as they undergo neither intestinal nor liver catabolism as none of these tissues contain the first key deamination enzyme (BCAT, reviewed below). Therefore, their systemic circulating concentrations reflect the administered or resorbed dose, respectively [12,13]. Moreover, leucine is among the most abundant AA in bodily proteins [14], which both predispose BCAAs to be the nutrient-sensing signals for many target tissues [15]. At the cellular level, BCAAs serve as (i) direct building blocks or nitrogen donors for proteosynthesis; (ii) energy/anaplerosis substrate while degraded eventually down to the final glucogenic (propionyl and succinyl-CoA) and ketogenic (acetyl-coA and acetoacetate) products and oxidized; or (iii) nutritional signals via mTOR activation. The intracellular pathways of BCAAs are summarized in Figure 1.

BCAAs share the first two katabolic enzymatic steps. The first is transamination catalyzed by branched-chain amino acid aminotransferase (BCAT) where deamination occurs and amino-residue is transferred to α-ketoglutarate to eventually form glutamate/glutamine. The product of the reaction is a keto-analogue of respective BCAAs, i.e., branched-chain fatty acid (BCFA), α-keto-isocaproate, α-keto-β-metylvalerate, or keto-isovalerate. A BCAT-catalyzed reaction is reversible and runs near equilibrium which means that the direction of the reaction depends on the availability of BCFA and nitrogen donors. Moreover, there exists an interorgan flux of branched carbon skeleton to prevent the loss of these essential compounds [16]. A further step towards eventual oxidation is an oxidative decarboxylation catalyzed by a branched-chain α-keto-dehydrogenase complex (BCKDH). The oxidative decarboxylation is an irreversible reaction that is highly regulated and coupled with NADH reduction. Products of decarboxylation are isovaleryl-CoA, isobutyryl-CoA and α-methyl-butyryl-CoA. These substrates consequently undergo a series of dehydrogenase mitochondrial reactions to eventually form degradation end-products acetoacetate, acetyl-CoA a propionyl-CoA. Importantly, at each catabolic step downstream from BCKDH, the oxidation intermediates may be used for anaplerosis, synthesis of FA or cholesterol [17].

BCAAs, namely leucine, signal to mTOR complex (mammalian/mechanistic target of rapamycin). This is an evolutionary-conserved nutrient-sensing serine–threonine kinase with two major complexes: mTORC1, which is responsible for cells’ anabolism to catabolism switch via control of metabolic state; and mTORC2, which downstream targets control cell survival, proliferation and cytoskeleton dynamics. The complex is an integrating system where many metabolic signals, both nutritional and hormonal, converge. Major phosphorylation targets of mTORC are p70S6 serine kinase 1 (S6K1) and 4EBP1 (binding protein 1 for eukaryotic initiation factor 4E) [15] that serve as regulators of transcriptional activity leading to many anabolic processes including protein synthesis, autophagy inhibition, and cell growth. Of note, phosphorylated S6K1 inactivates the insulin receptor substrate (IRS) and downregulates the insulin cascade activity, decreasing insulin-dependent substrate uptake in the time of the BCAA abundance [18].

## 3. Circulating BCAAs Associated with Insulin Resistance

Individual BCAAs share common structural resemblance and their circulating levels are synchronized in many clinical states, physical activity, fasting [19], and diabetes [17]. The association between the elevated circulating BCAAs and insulin resistance was described more than fifty years ago but it has just recently received a significant scientific acknowledgement [20,21]. Peripheral BCAA levels predict a risk of incident diabetes up to twelve years before its manifestation [22] and lifestyle intervention [20]. In line with this observation, bariatric surgery [23] that reverts metabolic syndrome decreases circulating BCAAs too. Whether there is any causal role in BCAA triggering IR remains to be confirmed. Nevertheless, a few plausible mechanisms of the increased BCAA availability interfering with insulin functions were described (Figure 2).

Firstly, the activation of downstream mediator of mTOR SK61 leads to the downregulation of insulin receptor substrate (IRS1 and IRS2) [9,18]. It was shown in vitro that BCAAs induce mTOR signaling in skeletal muscle reflected by a decreased glucose uptake [24]. Persistent mTOR activation due to a chronic BCAA overload may therefore contribute to insulin resistance [25].

Secondly, BCAA catabolic intermediates feed into other metabolic pathways, namely TCA, potentially disturbing fine-tuned equilibria dependent on substrate availability. This effect is also referred to as anaplerotic stress [26]. Several BCFA could cause mitochondrial dysfunction and inactivation of pyruvate dehydrogenase in the liver and heart. Once bound to CoA, BCFA can be only transported out of the mitochondria as acylcarnitines. Acylcarnitines are abundant in various organic acidurias and their levels correlate with functional impairment, namely mitochondrial dysfunction. As several BCAA-derived acylcarnitines are elevated in circulation of T2DM patients, it was suggested that this effect may link elevated BCAAs with mitochondrial dysfunction in metabolically active organs [27].

The association is far from being straightforward. It has been repeatedly shown that BCAAs exert insulinotropic effects via multiple mechanisms on beta cells: insulin synthesis activated by mTOR [28], acting on glutamate dehydrogenase [28], and stimulating incretin secretion [12]. Having such properties, it has been suggested that BCAA and/or diets rich in BCAAs could improve metabolic status in T2DM [29,30].

Moreover, BCAAs stimulate muscle protein synthesis and inhibit protein degradation [31]. It was shown that namely leucine and its derivates, when supplemented, can not only revert sarcopenia in patients with T2DM but also ameliorate their insulin sensitivity [32,33]. BCAAs were, hence, suggested as nutriceuticals to prevent lean mass loss [34,35].

Circulating levels of BCAAs depend on several factors modulating their appearance/disappearance. Rate of disappearance (R_d_) is determined by protein synthesis, urinary excretion, and catabolism/oxidation in tissues whereas the rate of appearance (R_a_) is determined by dietary intake, intestinal resorption, and tissue protein degradation. R_a_/R_d_ mismatch leading to elevated circulating BCAAs could be eventually caused by a shift in any of these.

*Factors decreasing R_d_*. Decreased protein synthesis, decreased oxidation, and increased excretion were all suggested as players in the elevated BCAA levels [25,36,37]. BCAAs constitute a significant portion of acyl-CoAs feeding lipogenesis in adipose tissue [38,39]. As it was shown, gene expression of BCKDH is down-regulated in adipose tissue of IR subjects; once the storage capacity is exceeded, there is a BCAA spill-over to the systemic circulation [40]. BCAAs stimulate protein synthesis [41] and are major building blocks and nitrogen donors for skeletal muscle. There is a postprandial flux of BCAAs into the skeletal muscle [42] increasing their R_d_. It is conceivable that decreased protein synthesis may be also linked to elevated BCAAs.

*Factors increasing R_a_*. As revised earlier, BCAAs stimulate protein synthesis whereas insulin blocks protein degradation. Loss of insulin action could lead to disinhibition of protein breakdown in the skeletal muscle and, therefore, to increased R_a_ of BCAAs [43]. As per increased dietary intake, BCAA intake does not contribute to fasting but to postprandial BCAA levels [12,25]. Anyhow, it may modulate other dependent factors as it was shown that dietary protein composition has the major impact on BCAA concentrations [44,45]. An important link between diet and bioavailability of BCAAs, i.e., the amount of digested and resorbed BCAAs, has emerged: gut microbiota composition and metabolic capacity [46,47,48].

## 4. Gut Microbiota as the Source of Amino Acids for the Host

Diet has been considered the main source of essential nutrients but recent research has unraveled the great importance of a “hidden metabolic organ”—the gut microbiota—in modulation of the availability of many necessary compounds to the host. Bacteria are capable of numerous biochemical processes, including digestion of proteins or peptides that escaped absorption in the upper part of the digestive tract [49,50] or AA synthesis from nonspecific nitrogen sources [51]. Therefore, the metabolic requirement of AAs may not be covered only by the diet but also by AAs synthesized de novo by the gastrointestinal microbiota [52]. This assumption was confirmed by experiments in which nitrogen-containing compounds (ammonium chloride, urea) labeled with ^15^N were provided in the diet and the label was later found in the AA pool of the host [53,54,55]. The final availability of the microbiota-derived AA to the host depends both on the macronutrient composition of the diet (i.e., diet rich in carbohydrates blocks protein fermentation) and the gut microbiota functional capacity. Consequently, the alterations in the gut microbiota composition, called dysbiosis, may substantially contribute to the metabolic deregulation of the host.

While identification of the bacterial species responsible for proteolytic fermentation was originally based on correlative methods and the culture on AAs containing media, the newer approach capitalize on Kyoto Encyclopedia of Genes and Genomes (KEGG) pathway analysis of annotated human gut bacterial genomes [56]. Unfortunately, all approaches face considerable limits.

Correlation studies comparing gut microbiota composition and feces/serum metabolome do not establish causality. Furthermore, as the small intestine seems to be responsible for a large part of the uptake of microbiota-derived AAs [57], composition of ileal microbiota is much more relevant, especially concerning microbial source of AAs. Nevertheless, most of the available studies describe composition of fecal microbiota that reflects rather the situation in the colon than in the small intestine. The fermentation studies are usually performed in simplified conditions while, in the real intestine, the availability of other diet-derived substrates can affect the metabolic activity of the intestinal microflora and thus also the products potentially available to the host. In the KEGG database, approx. 21% of the reactions remain unclassified, allowing only for a crude estimation of the gut microbiota functional capacity [58].

In spite of these limitations, several protein fermenters and amino acid producers have already been identified. In vitro experiments showed major protein fermenters: *Clostridium*, *Fusobacterium*, *Bacteroides*, *Actinomyces*, *Propionibacterium*, and *Peptostreptococci* [59]. The most abundant AA-fermenting bacteria in the human small intestine are bacteria belonging to the *Clostridium* clusters, the *Bacillus-Lactobacillus-Streptococcus* groups, and *Proteobacteria* [60]. In the large intestine of healthy humans, bacteria belonging to the *Clostridia* and *Peptostreptococci* appear to be the most prevalent species involved in AA fermentation [60,61,62]. BCFA abundance, i.e., the marker of BCAA fermentation, has also been correlated with decreased *Firmicutes* and increased unknown *Bacteroidetes*, as well as *Prevotella* spp., *Bacteroides ovatus*, *Bacteroides thetaiotamicron*, and *Clostridium* spp. in an artificial colon model of high-protein diets [63].

Gut bacterial proteins encompass a higher ratio of BCAAs to other amino acids compared with the mammalian organism [60]. They represent important nutrients in bacterial physiology and ensure multiple functions, including protein synthesis, nutrient signaling, adaptation to amino acid starvation, or regulation of virulence gene expression [64]. Bacteria synthesize BCAAs through conserved pathways that are present in fungi and plants but absent in mammals (Figure 3); these biosynthetic pathways also provide intermediates for the synthesis of BCFA [64]. SCFA and BCFA synthesis share the same multienzyme pathway; however, the substrates that initiate the pathway differ. Acetyl-CoA serves as the substrate for SCFA synthesis, whereas branched-chain acyl-CoA serves as the substrate for BCFA synthesis [65] (Zhang 2008). Depending on the metabolic requirements, BCFAs may be converted to BCAAs [66,67] (Sun 2020).

The actual metabolic program of bacteria significantly depends on available substrates. A landmark study by David et al. showed that short switch to an animal- or vegetable-based diet was associated with prompt alteration in fecal metabolome composition. A plant-based diet was associated with an increase in fecal SCFA while an animal-based diet led to the higher production of BCFAs (Figure 4) [68] (David). While literature dealing with the effect of dietary carbohydrates or fiber on bacterial fermentation program is quite rich, there is much less information available regarding microbial breakdown and fermentation of proteins. Based on the rather scarce evidence, it can be concluded that high protein intake is associated with an increase in *Bacteroides*, *Fusobacterium*, *Proteobacteria*, *Desulfovibrio*, *Bilophila wadsworthia*, *Clostridium*, *Ruminociccus*, *Eubacterium*, *A*. *putredinis* and *Bacteroides* sp., the last two genera positively correlating with fecal BCFAs. On the other hand, a decrease in *Firmicutes*, Archaea, *Megasphera*, *Selenomonas*, *Acidaminococcus*, *Bifidobacterium*, and *Prevotella* was observed. High-protein diets tend to lead to a decrease in the SCFA production, while BCAA and BCFA stool content is elevated [69].

The above mentioned findings support the hypothesis that microbiota-related components may be an important contributor to the diet-derived end-products available to the host.

## 5. Findings Derived from Observational Studies

### 5.1. Human Studies

Obesity, insulin resistance, and T2D are associated with elevated levels of circulating BCAA. In the landmark study involving 277 insulin-resistant, non-diabetic Danish subjects, Pedersen et al. demonstrated that the abundance of genes encoding enzymes involved in BCAA synthesis positively correlated with insulin resistance while the opposite relationship was observed in the case of genes encoding bacterial inward transporters for these AA [46]. Furthermore, they identified *Prevotella copri* and *Bacteroides vulgatus* as the main species responsible for the elevated BCAA biosynthesis and *Butyrivibrio crosstus* and *Eubacterium siraeum* for the decreased inward BCAA transport. Several other studies have reported altered microbiota composition and functional capacity for BCAA biosynthesis/degradation in both human and animal models of obesity or T2D since then. A metagenome-wide association study combined with serum metabolomic profiling performed in 118 obese and 105 lean healthy young Chinese individuals unraveled obesity-associated gut microbial signature linked to changes in circulating metabolites. Among other findings, the authors showed that microbiota of obese individuals have higher potential to produce aromatic and branched-chain AA whereas the BCAA degradation pathways were depleted [47]. In line with these findings, serum concentrations of Phe, Tyr, Leu, Ile, and Val were higher in obese than in lean subjects. These AA were inversely correlated with species from *Bacteroides* genus, including *B. thetaiotaomicron*, *B. intestinalis*, *B. ovatus*, and *B. uniformis.* The Malmö Offspring Study, encompassing 92 Swedish adults, revealed strong positive association between the concentration of serum BCAA and related metabolites such as BCFAs (3-methyl-2-oxovalerate, α-ketoisovalerate and α-ketoisocaproate) or short-chain acylcarnitines (isovalerylcarnitine) and BMI [70]. Furthermore, the strongest predictor of a high BMI in this study, glutamate, is a byproduct of the first step of BCAA oxidation where the amino group is transferred to α-ketoglutarate. Metabolites predictive of a higher BMI were correlated with four gut bacterial genera. In case of *Dorea*, *Blautia* and *Ruminococcus* (all belonging to the *Lachnoclostridiaceae* family) there was a positive correlation while in the case of the order *SHA-98* the relationship was inverse. The associations between the gut microbiota composition and circulating metabolites that may have an impact on cardiovascular or metabolic traits were addressed in METSIM study involving 531 Finnish males [71]. In this study, 40 significant associations between 17 traits and 23 unique bacterial operational taxonomic units (OTUs) were identified. In line with the study by Ottosson et al. [70], BCAAs were positively associated with the abundance of *Blautia*, a high BMI, and high homeostatic model assessment of insulin resistance (HOMA-IR) values. In addition, BCAA negatively correlated with *Christensenellaceae*. Wang et al. compared the gut microbiota and serum metabolome of relatively small groups of subjects following different dietary patterns, i.e., vegans (VG, *n* = 12), lacto-ovo-vegetarians (VEG, *n* = 12), and omnivores (OMNI, *n* = 12) [72]. Twenty-six and twenty-seven metabolites, including BCAAs, significantly contributed to the separation of OMNI from VEG and VG, respectively, with BCAA being higher in the OMNI group. The microbiome composition defined by the whole-genome sequencing of VG and VEG was significantly different from the OMNI group at the level of family, genus, and species. Interestingly, *P. copri* was identified by Pedersen et al. as the microorganism most positively associated with elevated serum BCAAs and was more abundant in the gut microbiota of the VG and VEG groups. On the other hand, metatranscriptomic analysis identified the BCAA degradation pathway as significantly enriched in VG and VEG compared with the OMNI group.

While the above-mentioned cross-sectional studies brought at least partly convergent outcomes, quite opposite results were obtained in a study of Mesnage describing metabolome and microbiome changes during a very different physiological situation, i.e., fasting and refeeding [73]. The study participants were examined at baseline, immediately after 10 days of fasting, after 4 days of refeeding and after 3 months of a habitual diet. The BCAA content dramatically increased (*p* = 0.0005) during fasting, returned to baseline after refeeding and declined significantly after 3 months. Regarding microbiome composition, fasting resulted in a decrease in *Lachnospiraceae* and *Ruminococcaceae* and a concomitant increase in the *Bacteroides* and *Proteobacteria* abundance. While *Lachnospiraceae* negatively correlated with the BCAA levels, *Bacteroidetes* correlated with them positively. This discrepancy points out the pitfalls of correlation studies and underlies the importance of the mechanistic approach. The outcomes from the above mentioned studies are summarized in Table 1.

### 5.2. Animal Studies

While it is difficult to adjust to a different dietary intake of individual AA in large human trials, animal studies allow for such a standardization and, therefore, for a more precise identification of microbiota-derived BCAA in the circulation. Several studies addressing the relationship between gut microbiota, circulating metabolites, and metabolic health were performed in animal models of obesity (mice fed a high-fat diet) [48,74] or diabetes (rats fed a high-fat diet and treated with streptozotocin) [75] (Table 2). In all models, serum BCAAs were significantly elevated and could be regarded as serum biomarkers of obesity-related insulin resistance. HFD induced significant changes in the gut microbiota composition characterized by the increased *Firmicutes* to *Bacteroidetes* ratio but, at the genus level, the outcomes were quite heterogeneous. Pathway analysis based on the metabolite abundances [48,75] or PICRUSt analysis based on the gut microbiota composition [74] identified an increased BCAA biosynthesis pathway in the HFD groups. The BCAA degradation pathway was decreased in HFD-fed mice only in the study of Zeng et al. [48] but not in the others.

## 6. Mechanistic Studies

### 6.1. The Effect of Bacterial Taxa Manipulation on the BCAA Metabolism

Numerous animal experiments demonstrated that manipulation of gut microbiota influences the systemic BCAA pool (Table 3). One type of experimental setting involved direct colonization of experimental animals with selected bacteria or the whole fecal microbiome. The first study in this field [76] demonstrated that ex-germ free mice humanized with fecal bacteria from obese individuals reproduced the obese phenotype associated with the elevated serum concentration of BCAAs. The PICRUSt analysis revealed the enrichment of AA-related pathways in the gut microbiome including valine, leucine, and isoleucine biosynthesis. As described above, *P. copri* was identified as the strongest driver species for the positive association between microbial BCAA synthesis in the gut and the traits of insulin resistance in a Danish cohort of non-diabetic males [46]. Aiming to confirm a causal relationship, Pedersen et al. performed an experiment in which conventionally raised HFD-fed mice, which were low in BCAAs, were repeatedly gavaged with *P. copri* for three weeks. Three weeks of *P. copri* challenge aggravated the glucose intolerance, increased the serum BCAA concentration, and reduced insulin sensitivity compared with sham-gavaged animals. Interestingly, *P. copri* represented only a minor component of the gut microbiome and it was also the only bacteria affected by the gavage during the whole experiment; this suggests that even low-abundant taxa may significantly affect the host metabolism. In another study, Zeng et al. [48] described the effect of gavaging conventionally raised HFD-fed mice with *Bacteroides ovatus*, i.e., bacteria that negatively correlated with circulating BCAA in human obesity [47]. Gavage with live, but not heat-killed *B. ovatus*, ameliorated diet-induced obesity, improved serum lipid profile, and significantly reduced the serum and fecal BCAA concentration compared with the sham-gavaged animals. The same effect was observed using *B. thetaiotaomicron* [47].

### 6.2. The Effect of Microbiota Modulation with Phytochemicals or Symbiotics on the BCAA Metabolism

A different strategy is based on the administration of compounds affecting the gut microbiome (Table 4). This manipulation is subsequently translated into serum AA, resp. BCAA composition as well as to the glucose metabolism-related parameters. To our knowledge, this effect was described after a treatment with citrus polymethoxyflavones (PMFE) [48], *Luffa cylindrical (luffa)* [74], berberine [77], glucomannans [75], and inulin [78]. The chemical nature of these compounds is quite heterogeneous. Inulin and glucomannan are indigestible polysaccharides (dietary fiber); berberine is a quaternary salt of benzylisoquinoline alkaloid; PMFE belongs to flavonoids; and *luffa* is a crude mixture of components including polyphenols, flavonoids, saponin, triterpenoids, and oleanolic acid. Their effect was tested in animal models of HFD-induced obesity and insulin resistance [48,74,77], diabetes [75] or in growing pigs [78]. All treatments alleviated HFD-induced obesity and ameliorated the glucose and/or lipid profiles. This effect was microbiota-dependent as it was partly or completely abolished by antibiotic treatment [74,77] or transferable by fecal transplantation from treated mice to HFD-fed ones [77]. The common effect of all interventions at the metabolome level was a decrease in serum BCAA concentrations. Regarding the gut microbiome composition, the effect of individual treatments is extremely variable but, in all studies, the authors reported altered functionality of the gut microbiota manifesting as either the downregulation of BCAA biosynthetic pathways [74,75], the upregulation of BCAA degradation pathways [48,78], or both [77]. The functional capacity of the gut microbiota was derived from its composition using PICRUSt [74,77], predicted by pathway analysis based on metabolome composition [48] or using both approaches [75,78]. Currently, there is only one available human study describing the effect of symbiotic (SG) administration (*Bifidobacterium lactis UBBLa-70* + fructooligosaccharide) in combination with low-energy intervention in obese women [79]. Both interventions led to a comparable weight loss but the serum BCAA concentration decreased only in the SG-supplemented group. Furthermore, a positive correlation for changes in Verrucomicrobia and isoleucine, as well as for *Firmicutes* and isoleucine, and total BCAAs were observed in the SG group. Sun et al. described the effect of the gentamicin intervention in mice infected with the Influenza virus. Besides the reduced survival of the mice, gentamicin treatment also resulted in an increased abundance of *Bacteroidetes*, decreased abundance of Proteobacteria, and elevated serum BCAA concentration [79].

## 7. Conclusions

Their unique biochemical and physiological characteristics predispose BCAAs to be the nutrient-sensing signals for many target tissues. BCAAs, namely leucine, signal to mTOR complex, an evolutionary conserved nutrient-sensing system ensuring optimal metabolic adaptation to nutrient availability and energy status. BCAAs play an essential role in physiologically switching off the insulin signaling; however, when present in excess, they are associated with insulin resistance and T2DM development. Being essential AAs, BCAAs cannot be synthesized and mammalian organism depends on their external sources. Besides the diet, the gut microbiota seem to be a substantial source of these nutrients. Here, we summarize the evidence for the gut microbiota being the player influencing the host circulating BCAA pool, thus potentially contributing to obesity and metabolic disorders. As per the mechanistic links, altered microbiome functional capacity manifested either by an increased abundance of BCAA biosynthesis and/or an decreased abundance of BCAA degradation pathways in the gut bacterial metagenome have been suggested. The data from microbiota-targeted intervention studies suggest that modulating the BCAA availability by affecting the gut microbiota may represent a promising therapeutic approach. Unfortunately, functional redundancy of microbial genomes is probably what currently precludes the identification of particular bacterial taxa responsible for BCAA modulating effects. Further research in this field is warranted in order to find the most effective ways to the prevention and treatment of insulin-resistant states.

## Figures and Tables

**Figure 1 biomolecules-11-01414-f001:**
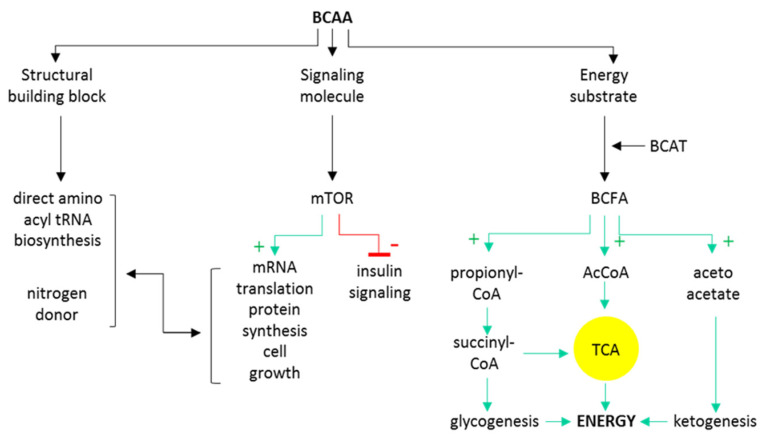
Schematic overview of major intracellular BCAA metabolic and signaling pathways. BCAT, branched-chain amino acid aminotransferase; BCFA, branched-chain fatty acids; AcCoA, acetyl coenzyme A; TCA, tricarboxylic acid cycle.

**Figure 2 biomolecules-11-01414-f002:**
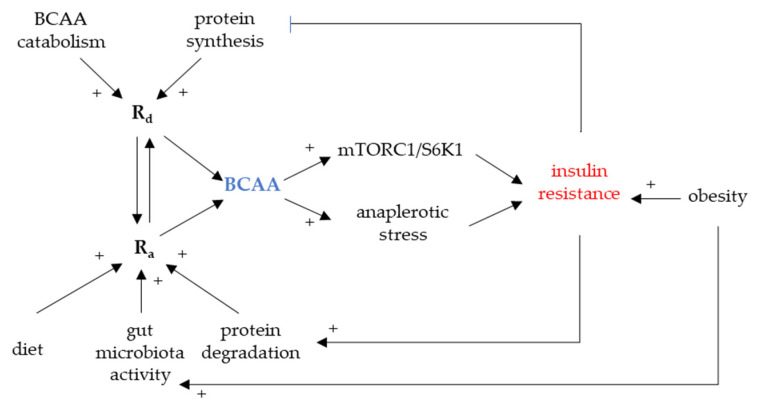
BCAA and insulin resistance development. mTORC1, mammalian/mechanistic target of rapamycin complex 1; R_a_, rate of appearance; R_d_, rate of disappearance; S6K1, p70S6 serine kinase 1.

**Figure 3 biomolecules-11-01414-f003:**
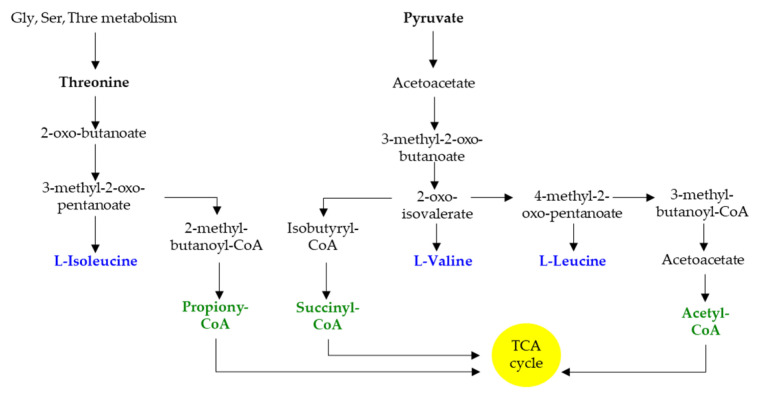
Schematic representation of BCAA and BCFA biosynthesis. Gly, glycine; Ser, serine; Thre, threonine.

**Figure 4 biomolecules-11-01414-f004:**
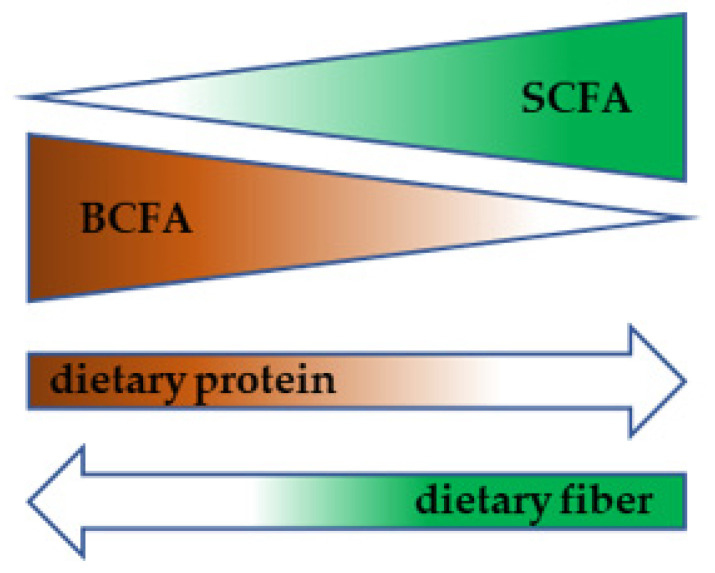
Diet-dependent metabolic switch in gut microbiota.

**Table 1 biomolecules-11-01414-t001:** Conditions associated with altered circulating BCAA concentration: human studies.

	Organism	Condition	Effect on Physiology	Microbiome Composition	Microbiome Functionality	Metabolome
Pedersen H. et al. [38]	human	IR/MetSy		increased *Prevotella. copri*, *B. vulgatus*; decreased *Butyrivibrio crosstus*, *Eubacterium siraeum*	upregulated BCAA biosynthesis (correlates with *P. copri* and *B. vulgatus*); downregulated inward BCAA transport (correlates with *B. crosstus* and *E. siraeum*)	increased s-BCAA
		T2D		increased *P. copri*, *B. vulgatus*; decreased *Butyrivibrio crosstus*, *Eubacterium siraeum*	Upregulated BCAA biosynthesis; downregulated inward BCAA transport.	increased s-BCAA
Ottosson F. et al. [56]	Human (Malmo Offspring Study)	obesity		increased *Dorea*, *Blautia*, *Ruminococcus*, positively correlate with PC-BMI decreased *SHA-98*, negatively correlate with PC-BMI		increased PC-BMI metabolites: glutamate, BCAA
Org E. et al. [57]	Human (METSIM study)	MetSy		*Blautia* positively correlated with s-BCAAs, high BMI and HOMA-IR *Christensenellaceae* negatively correlated with s-BCAAs		
Wang F. et al. [58]	human	VG VEG OMNI	VG and VEG vs OMNI: lower intake of energy, fat, chol.; higher intake of carbohydrates and fiber	VG and VEG vs OMNI: higher *Prevotella* (including *P. copri*) lower *Alistipes*, *Bacteroides* (NOT *B. thetaiotaomicron* or *B. ovatus*), *Bilophila*, *Collinsella*, *Parabacteroides*	VG and VEG vs OMNI: upregulated BCAAs degradation.	VG and VEG vs OMNI: lower s-BCAAs
Mesnage R. et al. [59]	human	10-day fasting (1046 kJ/day)	switch from carbohydrate to fatty acid oxidation, ketosis	decreased *Lachnospiraceae*, *Ruminococcaceae*; increased *E.coli*, *B. fragilis*, *B. thetaiotaomicron*, *Bilophila waldsworthia*		increased s-BCAA; negative correlation: BCAA/*Lachnospiraceae* positive correlation: BCAA/*B.fragilis*, *B. thetaiotaomicron*, *Bilophila*

BCAA, branched-chain amino acids; HFD, high-fat diet; HOMA-IR, homeostatic model assessment of insulin resistance, IR, insulin resistance; MetSy, metabolic syndrome; OMNI, omnivores; PC-BMI, metabolites predictive of BMI; s-BCAA, serum branched-chain amino acids; T2D, type 2 diabetes; VEG, vegetarians; VG, vegan.

**Table 2 biomolecules-11-01414-t002:** Conditions associated with altered circulating BCAA concentration: animal studies.

	Organism	Condition	Effect on Physiology	Microbiome Composition	Microbiome Functionality	Metabolome
Zeng et al. [40]	mice	HFD	weight gain; compromised glucose homeostasis and serum lipid profile; stimulated mTOR/p70S6K/SREB pathway	increased *Firmicutes to Bacteroidetes* ratio, *Ruminococcus*; decreased *S24-7*, *Ruminococcaceae*, *Lachnospiraceae*, *Bacteroides*, *Oscillospira*, *Rikenellaceae*	upregulated BCAA biosynthesis, decreased BCAA degradation	serum: shift in 73 metabolites feces: shift in 91 metabolites both including elevated BCAA
Zhang et al. [60]	mice	HFD	obesity-associated insulin resistance	increased *Enterorhabdus*, *Acetatifactor*, *Butyricoccus*, *Sterptococcus*, *Eubacterium_xylanophilum_group*, *Escherichia-Shigella*; decreased *Alloprevotella*, *Parasutterella*, *Parabacteroides*, *Eubacterium_coprostanoligenes_group*, *Christensenellaceae_R-7_group.*	upregulated BCAA synthesis; no change in BCAA degradation	increased s-BCAA
Chen et al. [61]	rat	diabetes model (HFD + STZ)	fasting hyperglycemia; decreased tissue BCAA metabolism	Increased Proteobacteria, *Lachnospiraceae_uncultured* and *Lachnoospiraceae_NK4A136* decreased *Firmicutes* to *Bacteroidetes* ratio, *Muribaculaceae*	Upregulated AA and BCAA biosynthesis; no change in BCAA degradation	altered AA metabolism, increased s-BCAA; positive correlation between *Bacteroides pectinophilus group*, *Bacteroides*, *Klebsiella*, *Prevotellaceae Ga6A1 group*, *Prevotellaceae NK3B31 group*, *Prevotellaceae UCG-001*, *Ruminiclostridium*, *Ruminiclostridium 1*, *5 and 9*, *Staphylococcus*, *Streptococcus* and BCAA

AA, amino acids; BCAA, branched-chain amino acids; HFD, high-fat diet; s-BCAA, serum branched-chain amino acids; STZ, streptozotocin.

**Table 3 biomolecules-11-01414-t003:** The effect of specific bacterial taxa on BCAAs concentration in serum.

	Model	Treatment	Effect on Physiology	Microbiome Composition	Microbiome Functionality	Metabolome
Ridaura et al. [62]	humanized mice	fecal Tx from obese twin	reproduction of obese phenotype		upregulated AA-metabolism related pathways (Phe, Lys, Leu, Ile, Val, Arg, Cys, Tyr)	increased s-BCAA, Met, Ser, Gly, Phe, Ala and Tyr
Pedersen H. et al. [38]	mice	HFD, *P. copri*/sham gavage	*P. copri* aggravated glucose tolerance, reduced insulin sensitivity	elevated *P. copri*; no other changes in microbiota composition	upregulated BCAA biosynthesis	increased s-BCAA
Zeng S. et al. [40]	mice	HFD + *B. ovatus*/sham gavage	decreased fat accumulation, ameliorated lipid profile and liver function tests	increased *B. ovatus*		decreased f- and s-BCAA
Liu R. et al. [39]	mice	HFD + *B. thetaiota-omicron*/sham gavage	lower adiposity; improved inflammatory status	increased *B. thetaiotaomicrone*, no substantial change in the whole microbiome		decreased circulating AA (glutamate, Phe, Leu, Val)

AA, amino acids; f-BCAA, fecal branched-chain amino acids; s-BCAA, serum branched-chain amino acids; Tx, transplantation.

**Table 4 biomolecules-11-01414-t004:** The effect of microbiota modulation with phytochemicals or symbiotics on BCAAs concentration in serum.

	Model	Treatment	Effect on Physiology	Microbiome Composition	Microbiome Functionality	Metabolome
Zeng S. et al. [40]	Mice HFD	PMFE	PMFE protected against MetSy in HFD mice PMFE inhibited mTOR/P70S6K/SREBP pathway	increased *Bacteroides*, *S24-7*, *Ruminococcaceae*, *Oscillospira*, *Lachnospiraceae*; decreased *Paraprevotella*, *Streptococcus*	upregulated BCAA degradation	prevention of HFD-induced increase in s-BCAA and f-BCAA
	fermentation in vitro	fresh feces (HFD mice) + PMFE added in vitro		increased *B. ovatus*, *B. thetaiotaomicron*, *B. vulgatus*, *B. caccae*, *B. stercoris*, *B. uniformis* decreased *B. fragilis*, *B. finegoldii*, *B. coprophilus*		
Zhang L. et al. [60]	Mice HFD	*Luffa cylindrica*	alleviation of obesity-associated insulin resistance	HFD induced significant shift in microbiota composition, *luffa* induced decrease in *Enterorhabdus*, *Butyricoccus*, *Eubacterium_xylanophilum_group*	downregulated BCAA synthesis; no change in BCAA degradation	decreased s-BCAA
		*Luffa cylindrica* + ATB	no effect on obesity-related parameters; partial alleviation of glucose intolerance	microbiota depletion		no effect on s-BCAA compared with HFD-fed mice
Yue S. et al. [63]	Mice HFD	berberine	alleviation of HFD-induced obesity and glucose intolerance	restoration of HFD-induced shift in *Firmicutes* to *Bacteroidetes* ratio increased *Akkermansia*, selective reductions in *Clostridiales*, *Streptococcaceae*, *Clostridiaceae*, *Prevotellaceae*, decreased *Streptococcus*, *Prevotella*	downregulated BCAA biosynthesis; upregulated BCAA degradation	prevented HFD-induced increase in s-BCAA (valine *p* < 0.05, leucine *p* = 0.17, isoleucine *p* = 0.06); downregulated bacterial taxa correlate positively with s-BCAA
Chen H. et al. [61]	rat HFD + streptozotocin	gluco-mannans	hypoglycemic, hypolipidemic, kidney-protective effect	decreased *Clostridium* spp., *Bacteroides* spp., *Prevotella* spp., *Klebsiella* spp., *Escherichia coli*, *Streptococcus* spp., *Staphylococcus aureus*	downregulated BCAA biosynthesis; no change in BCAA degradation	decreased s-BCAA
Wu W. et al. [64]	growing pig	inulin	lower s-cholesterol and s-glucose	increased abundance of 10 genera (including *Prevotella* and *Succinivibrio*), decreased abundance of six genera	upregulated BCAA degradation	decreased s-BCAA positive correlations between *Prevotella*/Val and Leu *Succinivibrio*/ValClostridium_sensu_stricto_1/Ile and Leu
Crovesy L. et al. [65]	obese women	low-energy diet				decreased glycerol; increased Arg, Glu and 2-oxoisovalerate
		low-energy diet + SG				dtto + increased Pyr, Ala; decreased citrate, Ile and total BCAA; positive correlations∆ Verrucomicrobia/∆ Ile∆ *Firmicutes*/∆ Ile and ∆ total BCAA
Sun Y. et al. [66]	mice influenza virus infection	gentamicin	reduced survival; decreased number of suppressor CD11b + Ly6G + cells, increased CD8 + cells in lung, enhanced inflammation	increased *Bacteroidetes* decreased *Proteobacteria*		increased s-BCAA

ATB, antibiotics; BCAA, branched-chain amino acids; f-BCAA, fecal branched-chain amino acids; HFD, high-fat diet; MetSy, metabolic syndrome; PMFE, citrus polymethoxyflavones; s-BCAA, serum branched-chain amino acids; SG, symbiotic (*Bifidobacterium lactis UBBLa-70* + fructooligosaccharide); ∆, the change in metabolite or phylum post vs pre intervention.

## Data Availability

Not applicable.

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
