# Peer review of "Gut Microbiota as the Link between Elevated BCAA Serum Levels and Insulin Resistance"

_biomolecules, 2021, doi:10.3390/biom11101414_

Round 1

Reviewer 1 Report

The Manuscript “Gut microbiota as the link between elevated BCAA serum levels and insulin resistance” by Gojda and Cahova is interesting and sound. It touches every aspect branched chain amino acids (BCAA) from synthesis to its metabolism and functional importance. The dietary role on BCAA and its implication of BCAA in insulin resistance in relation to gut microbiota is discussed very well.

Author Response

We thank the reviewer for positive evaluation of our manuscript. 

Reviewer 2 Report

The manuscript submitted for publication in Biomolecules MDPI, by Gojda and Cahova titled: “Gut microbiota as the link between elevated BCAA serum levels and insulin resistance” is a review aiming to discuss the evidence on the association and links between Branched Chain Amino Acids (BCAA) and insulin resistance and how that axis is affected by the gut microbiome. The topic is of interest as the mechanistic relationships and systems governing these associations are unclear and there is significant interest from the scientific community regarding the topic.

The manuscript is well written and organized overall. There is good flow, and it is easy to read through and follow. In general, the points are clearly presented, and the tables are useful can be followed easily providing a summarized version on the studies cited.

The reviewer would like to offer the following points aiming at the improvement of the manuscript.

  1. One high-level conceptual point is the integration of the work presented. It would be most powerful and useful to the reader to provide a section at the end discussing the current state of knowledge and the commonalities in terms of outcomes, observations and finally conclusions among studies. In short it would be useful to put the essentials together and discuss the work in its entirety after the sectioned presentation of the studies and data. A couple of useful reviews that the authors can consider using:
  • Maykish A, Sikalidis AK. Utilization of Hydroxyl-Methyl Butyrate, Leucine, Glutamine and Arginine Supplementation in Nutritional Management of Sarcopenia-Implications and Clinical Considerations for Type 2 Diabetes Mellitus Risk Modulation. J Pers Med. 2020 Mar 24;10(1):19. doi: 10.3390/jpm10010019. PMID: 32213854; PMCID: PMC7151606.
  • Sikalidis AK, Maykish A. The Gut Microbiome and Type 2 Diabetes Mellitus: Discussing a Complex Relationship. Biomedicines. 2020 Jan 7;8(1):8. doi: 10.3390/biomedicines8010008. PMID: 31936158; PMCID: PMC7168169.
  1. Given the fact that the number of studies cited and used in the tables is not that high it becomes more challenging to draw solid conclusions. Therefore a more comprehensive discussion on the mechanisms is warranted and that is why the reviewer is suggesting the aforementioned publications and is recommending the addition of the section discussed in his earlier point.
  2. A conceptual pictogram constructing the gist of the manuscript would be useful and could also be used as a graphical abstract.

Author Response

Response to Reviewer 2

We thank the reviewer for the time and inspiring comments that help us to improve the manuscript. We made appropriate changes in the text as shown below. We believe that we have addressed all of the issues raised and have provided complete replies to this reviewer's comments.

Comments

  1. One high-level conceptual point is the integration of the work presented. It would be most powerful and useful to the reader to provide a section at the end discussing the current state of knowledge and the commonalities in terms of outcomes, observations and finally conclusions among studies. In short it would be useful to put the essentials together and discuss the work in its entirety after the sectioned presentation of the studies and data.

We significantly re-wrote the Conclusions (lines 424-444) aiming to summarize the outcomes of the studies discussed earlier in the text.

  1. Given the fact that the number of studies cited and used in the tables is not that high it becomes more challenging to draw solid conclusions. Therefore, a more comprehensive discussion on the mechanisms is warranted and that is why the reviewer is suggesting the aforementioned publications and is recommending the addition of the section discussed in his earlier point.

We added a new paragraph discussing the potentially positive role of BCAA in prevention/reversal of insulin resistance in specific physiological states, i.e. sarcopenia (lines 142 – 150). We thank the reviewer for reminding us about the interesting studies, which were incorporated into our review.

  1. A conceptual pictogram constructing the gist of the manuscript would be useful and could also be used as a graphical abstract.

We added required pictogram (Figure 3).

Reviewer 3 Report

  1. The authors have enumerated the crosslink between BCAA and insulin resistance with gut microbiota as mediator.
  2. The review on the whole is informative.
  3. I strongly believe that the sections "Branched chain amino acids are both nutrients and signaling molecules" and Gut microbiota as the source of amino acids for the host require mechanistic pathway figures. 
  4. There are a few areas which require language corrections.
  5. Additionally, section 4 needs a little more evidences to support. 
  6. I feel that illustrations will make this review more attractive.

Author Response

Response to Reviewer 3

We thank the reviewer for her/his comment and we believe that responding to this reviewer´s point has led to considerable improvement of our manuscript.

Comments

  1. The authors have enumerated the crosslink between BCAA and insulin resistance with gut microbiota as mediator.
  2. The review on the whole is informative.
  3. I strongly believe that the sections "Branched chain amino acids are both nutrients and signaling molecules" and Gut microbiota as the source of amino acids for the host require mechanistic pathway figures. 

Figure 3 (Schematic representation of BCAA and BCFA biosynthesis) and Figure 4 (Diet dependent metabolic switch in gut microbiota) were added.

4. There are a few areas which require language corrections.

The language revision was performed.

5. Additionally, section 4 needs a little more evidences to support. 

Section 4 was complemented with two paragraphs and two figures describing the pathways involved in BCAA biosynthesis and the interrelationship between gut microbiota metabolic program and diet.

6. I feel that illustrations will make this review more attractive.

Three more figures were added.

Round 2

Reviewer 2 Report

The authors have addressed the reviewer's points reasonably adequately.

Reviewer 3 Report

The present manuscript version is good to go ahead.